# BoxCD: Leveraging Contrastive Probabilistic Box Embedding for Effective and Efficient Learner Modeling

## Abstract

In digital education, Cognitive Diagnosis (CD) is essential for modeling learners' cognitive states, such as problem-solving ability and knowledge proficiency, by analyzing their response data, like answer correctness. However, traditional CD methods struggle with *effectiveness* and *efficiency*. They fail to capture the diversity and uncertainty of learners' cognitive states. Additionally, response prediction can be time-consuming. To address these issues, we propose BoxCD, a contrastive probabilistic box embedding model for cognitive diagnosis. BoxCD utilizes high-dimensional axis-aligned hyper-rectangles (boxes) to represent learners and exercises, with the volume of intersecting boxes used to predict learners' responses. This approach effectively captures semantic diversity and uncertainty while enhancing diagnostic effectiveness. To stabilize box embeddings, we integrate contrastive learning objectives with response prediction goals, optimizing the distance between positive and negative samples of learner and exercise boxes to improve uniformity. Additionally, we develop a rank-based response prediction method that leverages the geometric properties of box embeddings to efficiently assess learners' response correctness. Comprehensive experiments on two real-world datasets demonstrate that BoxCD outperforms traditional CD models in both effectiveness and efficiency, showcasing its potential to enhance personalized learning in digital education platforms.

**ACM Reference Format:**
Anonymous Author(s). 2024. BoxCD: Leveraging Contrastive Probabilistic Box Embedding for Effective and Efficient Learner Modeling. In . ACM, New York, NY, USA, 10 pages. https://doi.org/10.1145/nnnnnnn.nnnnnnn

## 1 Introduction

Digital education platforms such as *Coursera.com* offer a wealth of learning resources, such as exercises, within a flexible online environment. This convenience attracts an increasing number of learners from diverse fields, such as law, engineering, and academia [16]. As online learning expands, there is a growing need for effective tools to assess learners and support personalized learning. A key activity in online learning is "practice", where learners independently select and complete exercises. By analyzing learners' response data (e.g., correctness of answers), Cognitive Diagnosis (CD) models can evaluate their cognitive states, such as problem-solving ability [10] or proficiency in specific knowledge concepts [33]. For instance, a CD model may diagnose a learner's mastery probability of the mathematical concept *function* as 0.7. The results of CD assessments facilitate personalized services, including exercise recommendations [13] and adaptive testing [32, 40]. Thus, research on CD for accurately assessing learners is of significant importance.

Since directly measuring learners' cognitive states is challenging, mainstream CD approaches obtain them indirectly [28]. They represent both learners' cognitive states and the features of practiced exercises (e.g., *difficulty*) as trainable vectors [5, 8], as illustrated in

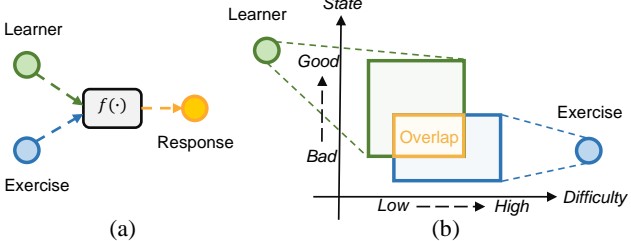

**Figure 1: Modeling framework for CD. (a) Learners and exercises are represented as vectors (points). (b) In BoxCD, these vector representations are transformed into box embeddings.**

Figure 1 (a). These vectors are optimized together by fitting learners' observed responses using a diagnosis function $f(\cdot)$. While effective, existing CD methods still face challenges in terms of **effectiveness** and **efficiency**. Regarding effectiveness, current vectorized representations of learners and exercises inadequately capture their diversity and uncertainty. For instance, a learner's cognitive state and an exercise's features fluctuate within specific ranges depending on the context. In a formal testing environment, stress and anxiety can impair performance, while learners may excel in daily practice due to reduced pressure. Consequently, the difficulty of exercises and other characteristics may also vary due to changes in learning states. Modeling such semantic diversity and uncertainty using single points in vector space is insufficient. In terms of efficiency, current methods predict the probability of a learner correctly answering an exercise using neural networks [8], dot products [24], or logistic-like functions [10]. These calculations, primarily involving neural networks, are time-consuming, especially when applied to large volumes of exercises in real-world educational platforms. It renders them unsuitable for rapid online services like adaptive testing [40]. Although some platforms use offline computation and store responses for timely online retrieval, the initial computation for each trained CD model is still time-consuming. In summary, there is a pressing need for a more comprehensive solution to enhance the effectiveness and efficiency of CD tasks.

Recently, geometric embedding techniques, such as probabilistic box embeddings [6, 23], have shown promise in addressing the current limitations of CD. Probabilistic box embeddings represent objects (e.g., learners and exercises) as high-dimensional axis-aligned hyper-rectangles. The interactions between these objects, such as the probability of a learner correctly answering an exercise, are quantified by the volume of their intersecting boxes. As shown in Figure 1 (b), mapping learners and exercises into box representations allows for a natural modeling of their diversity and uncertainty. Moreover, it becomes straightforward to determine whether the learner boxes and exercise boxes overlap in space. This property enables us to efficiently identify which exercises a learner can answer correctly, thereby intuitively reducing the time required for response predictions across numerous exercises.

However, integrating probabilistic box embeddings into CD models presents several technical challenges: (1) **Stabilizing Box Embeddings.** Learners' online learning can be irregular, often focusing only on problems they excel at, leading to sparse response records for most learners [38]. Box embeddings optimized on sparse records are prone to instability [23]. Furthermore, compared to traditional vector embeddings, box embeddings optimized by calculating intersecting volumes are more susceptible to overlap [18]. This overlap can hinder the differentiation of learners' and exercises' representations, counteracting the goal of intelligent education to distinguish between various learner types and the interactions between learners and exercises. To address this, we propose combining contrastive learning objectives to enhance the uniformity of box representations by bringing positive pairs closer together and separating negative pairs. However, this approach is limited by the second challenge: (2) **Training Dilemma from Disjoint Boxes.** When the learner and exercise boxes are disjoint, the gradient from the vanilla training loss of CD (i.e., predicting responses based on box intersections) vanishes, as shown by [6]. Similarly, for a pair of separated contrast training samples, the contrastive learning loss does not provide gradients for further movement. (3) **High Efficiency in Response Prediction.** While assessing the correctness of learners' responses based on box overlap may seem straightforward visually, formalizing this useful prior mathematically and integrating it into the CD model remains an unresolved issue.

To address these three limitations, we propose a contrastive probabilistic *Box* embedding model for *C*ognitive *D*iagnosis (*BoxCD*) to achieve an effective and efficient learner modeling. By utilizing probabilistic box embeddings, we can better represent learners and exercises in cognitive diagnosis tasks. The volume of overlap between learner and exercise boxes serves as the basis for response predictions. This method offers satisfactory psychological interpretability within the context of CD. To tackle the first limitation, BoxCD combines contrastive learning objectives with the intrinsic response prediction goal of CD, optimizing the distance between positive and negative samples of learner and exercise boxes. Figure 4 illustrates that the distribution of learner and exercise boxes becomes more uniform after applying contrastive learning. To address the second limitation, we employ a Gumbel-based volume calculation objective [6] to prevent gradient vanishing. After learning the box embeddings, we implement a rank-based response prediction method using box intersections to quickly determine whether each learner can answer the exercises correctly. Since the probability of answering incorrectly for unpracticed exercises is zero, there is no need to predict the performance for such cases. Consequently, this narrows the scope of exercises for which the probability of answering correctly needs further prediction, thereby improving efficiency. Comprehensive experimental results on two real-world datasets demonstrate that the proposed BoxCD outperforms traditional CD models with vector embeddings in both effectiveness and efficiency.

## 2 Related Work

### 2.1 Cognitive Diagnosis

As a fundamental task, cognitive diagnosis (CD) has been extensively studied in educational psychology for decades [2, 20]. Its primary aim is to profile learners' implicit cognitive states, such as their abilities or proficiency in specific knowledge concepts, by analyzing observed practice records (e.g., correct and incorrect responses). Existing research on CD operates under the assumption that learners' knowledge proficiency correlates with their practice performance, following the psychological Monotonicity assumption [33]. Consequently, diagnosis is achieved by predicting learners' practice responses [8]. The diagnostic results of CD can be applied to various intelligent applications, including exercise recommendation [14] and adaptive testing [32], prompting the development of numerous CD models in recent years. Early studies, such as IRT [10] and MIRT [1], as well as matrix factorization approaches like MCD [24], focus on modeling learners' answering processes by predicting the probability of correct responses, utilizing latent factors to represent learners' abilities. However, these methods often lack interpretability, as they cannot provide explicit multidimensional diagnostic results for each knowledge concept. To enhance interpretability, subsequent CD models have aimed to incorporate knowledge concepts related to questions, allowing for a diagnosis of learners' proficiency across all knowledge concepts [2, 5, 8, 22, 26, 30, 36–38]. NCDM [33], one of the most representative models, employs neural networks to capture complex interactions, moving beyond the linear interaction functions used in earlier works (e.g., IRT and MIRT).

In summary, existing CD studies represent learners' cognitive states through trainable vectors. However, as discussed in our introduction, this approach has limitations regarding both effectiveness and efficiency.

### 2.2 Probabilistic Box Embedding

Probabilistic box embeddings [29, 31] have been developed to model objects as high-dimensional, axis-aligned hyperrectangles. These box embeddings exhibit strong representational capabilities, especially for transitive relations. However, optimizing them using standard gradient descent techniques presents significant challenges. To address this, [17] employs Gaussian convolution to smooth the edges of the boxes, effectively alleviating the zero gradient problem. Additionally, [6] utilizes the Gumbel distribution to tackle local identifiability issues.

Recently, several applications based on box representations have emerged. For example, Query2Box [27] leverages box embeddings for logical reasoning within knowledge graphs, encoding queries and entities as boxes. Other studies [4, 18, 19, 23] aim to capture user interests by examining the intersections of items users have interacted with for recommendation tasks. However, to the best of our knowledge, research on the application of box embeddings in educational contexts remains unexplored.

## 3 Background

In this section, we first demonstrate the basic setup of Cognitive Diagnosis (CD). Then, we briefly introduce how the previous attempts model the CD task with vector embeddings. Finally, we define the key notions and operations of box embeddings that will be used to implement the BoxCD model.

## 3.1 Basic Setup of CD

***Notions.*** In a CD model, there are $N$ learners $\mathcal{U} = \{u_1, u_2, \ldots, u_N\}$, $M$ exercises $\mathcal{E} = \{e_1, e_2, \ldots, e_M\}$ and $C$ knowledge concepts. Each learner $u_i$'s cognitive state (e.g., problem-solving ability or knowledge proficiency) and each exercise $e_j$'s features (e.g., difficulty) are represented as trainable embeddings such as vector embeddings or box embeddings. Each exercise tests one or more of the $C$ knowledge concepts. The responses of the learners are provided in triples $\mathcal{R} = \{(u_i, e_j, y_{i,j})\}$, where $y_{i,j}$ (either 1 or 0, as training label) indicates whether the learner $u_i$ answered exercise $e_j$ correctly. Overall, the input data for training a CD model includes each response data $R_{i,j} = (u_i, e_j, y_{i,j}) \in \mathcal{R}$, as well as corresponding vector or box embeddings of learner $u_i$ and exercise $e_j$.

Given the above input, the **goal** of a CD model $f(\cdot)$ is to: (1) infer the cognitive state of each learner, and (2) predict learners' responses to unpracticed exercises.

***Optimization.*** Since directly obtaining learners' true cognitive states as training labels is challenging [2], existing CD models optimize these states indirectly by fitting learners' responses to specific exercises (i.e., whether they answer correctly) based on observed response data. Through joint training, these models can optimize learners' abilities or proficiency on specific knowledge concepts, as indicated by the exercises they have practiced. Additionally, they can derive meaningful exercise features, such as *difficulty*.

To ensure the interpretability of diagnostic results, CD models adhere to the psychological **Monotonicity** assumption [34], which posits that the probability of a correct response increases with the learner's cognitive state.

## 3.2 Modeling CD with Vector Embedding

In vector embedding setups, the cognitive state of each learner $u_i \in \mathcal{U}$ and the feature of each exercise $e_j \in \mathcal{E}$ are represented as $d$-dimensional vectors, $\mathbf{u}_i$ and $\mathbf{e}_j$, respectively. For ability-focused models, $d = 1$ for single-aspect models such as IRT [10], and $d \in \mathbb{R}^+$ for multi-aspect models such as MIRT [1]. For proficiency-focused models such as NCDM [33] and RCD [8], $d$ always equals the number of knowledge concepts $C$, where $u_{i,c}$ indicates the mastery probability of learner $i$ on concept $c$ tested by exercise $j$. After training by fitting learner responses, vectors of learner cognitive states and exercise features, and parameters of the CD model $f(\cdot)$ are jointly optimized.

To meet the **Monotonicity** assumption, diagnosis function $f(\cdot)$ should be monotonically increasing (e.g., Sigmoid) or be the neural network with non-negative weights, ensuring $\partial f(\cdot)/\partial \mathbf{u}_i \geq 0$.

## 3.3 Probabilistic Box Embedding

***Notions.*** In probabilistic box embeddings [6, 23], given an object $x$ (e.g., the learner or exercise in our context), a $d$-dimensional box embedding (i.e., an axis-aligned hyper-rectangle) is used to represent it, in which the parameters contain two vectors that correspond to the lower and upper boundaries of the box in $d$ dimensions, i.e., $\mathbf{x}^\wedge$ and $\mathbf{x}^\vee$, respectively. Let $box(x)$ associate the box embedding of object $x$, and we have

$$box(x) = \langle \mathbf{x}^\wedge, \mathbf{x}^\vee \rangle = \langle [x_1^\wedge, x_1^\vee], [x_2^\wedge, x_2^\vee], \ldots, [x_d^\wedge, x_d^\vee] \rangle \in \mathbb{R}^1. \tag{1}$$

The the volume of $box(x)$ is the interval lengths of the $d$-dimensional boundaries as follows:

$$V(box(x)) = \prod_{k=1}^{d} \left( x_k^\vee - x_k^\wedge \right) \in \mathbb{R}^1. \tag{2}$$

Below, we introduce two existing box operations.

***Intersection of Two Boxes.*** Given the box representations $box(a)$ and $box(b)$ of any two objects $a$ and $b$, we can obtain their overlapping region, a $d$-dimensional box, $box(a) \cap box(b)$ by intersection:

$$box(a) \cap box(b) = \langle [a_1 \cap b_1], [a_2 \cap b_2], \ldots, [a_d \cap b_d] \rangle, \tag{3}$$

where the $k$-dimensional lower and upper boundaries of $box(a) \cap box(b)$ are calculated by $a_k \cap b_k = \left[ \max\left(a_k^\wedge, b_k^\wedge\right), \min\left(a_k^\vee, b_k^\vee\right) \right]$. If two boxes are disjoint, it means there always exists at least one dimension $k$ such that $\max\left(a_k^\wedge, b_k^\wedge\right) > \min\left(a_k^\vee, b_k^\vee\right)$.

The volume of $box(a) \cap box(b)$ is calculated by:

$$V(box(a) \cap box(b)) = \prod_{k=1}^{d} \max\left(0, \min\left(a_k^\vee, b_k^\vee\right) - \max\left(a_k^\wedge, b_k^\wedge\right)\right) \in \mathbb{R}^1. \tag{4}$$

The Eq. (4) ensures the volume of $V(box(a) \cap box(b))$ is always non-negative, even if $\min\left(a_k^\vee, b_k^\vee\right)$ might be smaller than $\max\left(a_k^\wedge, b_k^\wedge\right)$.

***Union of Multiple Boxes.*** Given a set of box representations $box(x_1), box(x_2), \ldots, box(x_n)$ of multiple objects $x_1, x_2, \ldots, x_n$, we can obtain their union region, a $d$-dimensional box, $box(x_1) \cup box(x_2) \cup \ldots \cup box(x_n)$ by union:

$$box(x_1) \cup box(x_2) \cup \ldots \cup box(x_n)$$
$$= \langle [x_{1,1} \cup x_{2,1} \cup \ldots \cup x_{n,1}], [x_{1,2} \cup x_{2,2} \cup \ldots \cup x_{n,2}], \tag{5}$$
$$\ldots, [x_{1,d} \cup x_{2,d} \cup \ldots \cup x_{n,d}] \rangle \in \mathbb{R}^d,$$

where the $k$-dimensional lower and upper boundaries of $box(x_{1,k}) \cup box(x_{2,k}) \cup \ldots \cup box(x_{n,k})$ are calculated by $x_{1,k} \cup x_{2,k} \cup \ldots \cup x_{n,k} = \left[ \min\left(x_{1,k}^\wedge, x_{2,k}^\wedge, \ldots, x_{n,k}^\wedge\right), \max\left(x_{1,k}^\vee, x_{2,k}^\vee, \ldots, x_{n,k}^\vee\right) \right]$. The union operation ensures that the boundaries span the entire region covered by all the boxes in each dimension.

The volume of $n$ boxes' union is calculated by:

$$V(box(x_1) \cup box(x_2) \cup \ldots \cup box(x_n))$$
$$= \prod_{k=1}^{d} \left( \max\left(x_{1,k}^\vee, x_{2,k}^\vee, \ldots, x_{n,k}^\vee\right) - \min\left(x_{1,k}^\wedge, x_{2,k}^\wedge, \ldots, x_{n,k}^\wedge\right) \right) \in \mathbb{R}^1. \tag{6}$$

The above introduction lays the foundation for exploring BoxCD in the context of box embeddings in § 4.1.

## 4 BoxCD Model

In this section, we first give the basic formulation and optimization of BoxCD (see § 4.1). Then, we introduce an additional contrastive box learning objective in the optimization process of BoxCD (see § 4.2), which addresses the first technical challenge by enhancing the discrimination of box representations. Afterwards, a Gumbel-based volume objective [6] is adopted to mitigate the second challenge of the gradient vanishing issue.

## 4.1 Formulation of BoxCD

In BoxCD, each learner $u_i \in \mathcal{U}$ and each exercise $e_j \in \mathcal{E}$ are represented as $d$-dimensional box embeddings, denoted as $box(u_i)$ and $box(e_j)$, respectively. Specifically, we have:

$$box(u_i) = \left\langle \mathbf{u}^{i,\wedge}, \mathbf{u}^{i,\vee} \right\rangle = \left\langle \left[ u_1^{i,\wedge}, u_1^{i,\vee} \right], \left[ u_2^{i,\wedge}, u_2^{i,\vee} \right], \ldots, \left[ u_d^{i,\wedge}, u_d^{i,\vee} \right] \right\rangle,$$
$$box(e_j) = \left\langle \mathbf{e}^{j,\wedge}, \mathbf{e}^{j,\vee} \right\rangle = \left\langle \left[ e_1^{j,\wedge}, e_1^{j,\vee} \right], \left[ e_2^{j,\wedge}, e_2^{j,\vee} \right], \ldots, \left[ e_d^{j,\wedge}, e_d^{j,\vee} \right] \right\rangle.$$
$$(7)$$

For the CD task, given the box embeddings of the learner and the exercise, it needs to predict the probability $\hat{y}_{i,j}$ that learner $u_i$ answers exercise $e_j$ correctly, the same as vector embedding-based CD. However, it is intractable to still apply neural networks, dot product or logistic-like functions, used in vector-based CD models, to predict response due to the complex structure within the box representations. Instead, we determine the predicted probability $\hat{y}_{i,j}$ by the volume of the intersection of their respective boxes,

$$\hat{y}_{i,j} = Sigmoid\left( V\left( box(u_i) \cap box(e_j) \right) \right), \quad (8)$$

where $Sigmoid(\cdot)$ is the Sigmoid function $Sigmoid(x) = 1/(1 + e^{-x})$, mapping the overlapping volume to a range of 0 to 1.

It is worth noting that, the intersection-based prediction (Eq. (8)) has the following spotlights: (**S1: cognitive diagnosis-oriented**) The intersection of the learner's box and the exercise's box serves as a reflection of the learner's response to the exercise, which aligns with the intrinsic training objective of cognitive diagnosis. (**S2: psychological interpretability**) This equation upholds the Monotonicity assumption commonly foundational in traditional CD models (as discussed in § (3.1)). The volume of the overlapping boxes between the learner and the exercise is monotonically proportional to the region of the learner's box, continuing until the learner's box completely encompasses the exercise box. This characteristic ensures psychological interpretability.

To optimize BoxCD, the predicted probability $\hat{y}_{i,j} \in (0, 1)$ is required to closely match the true response $y_{i,j} \in \{0, 1\}$. We adopt the binary cross-entropy loss as the optimization objective, which is defined as:

$$\mathcal{L}_{i,j}^{res} = -y_{i,j} \log(\hat{y}_{i,j}) - (1 - y_{i,j}) \log(1 - \hat{y}_{i,j}). \quad (9)$$

## 4.2 Contrastive Box Learning Objective

To address the first challenge of stabilizing box embeddings, we incorporate two contrastive learning objectives into the BoxCD training process. The **contrastive learner-learner objective** aims to pull similar learner box representations closer together while pushing dissimilar ones further apart, thereby facilitating the learning of discriminative cognitive states among learners. Meanwhile, the **contrastive learner-exercise objective** aligns learner box representations with the exercise boxes they can correctly answer, while distancing them from exercises they cannot solve or have not yet practiced. This approach enhances response prediction through learner and exercise box intersection operations. As a result, the learned box embeddings become both stable and distinguishable.

Given a batch of training data, denoted as $\mathcal{R}^b$, each entry corresponds to a response record $(u_i, e_j, y_{i,j}) \in \mathcal{R}^b$. For constructing two contrastive learning objectives, we pair each learner in the batch with both positive and negative samples, including learners and exercise samples.

***Contrastive Learner-learner Objective.*** For a learner $u_i$, we define the positive learner samples as the $p$ most similar learners in the batch, while the negative learner samples consist of the $q$ least similar learners. To compute the similarity between learners, we employ a straightforward operation widely used in prior research [21], which involves calculating similarity scores based on their response records. Specifically, we represent each learner $u_i$ by an $M$-dimensional vector $\mathbf{r}$, where $r_{u,j}$ takes on values of 1, 0, or -1, indicating whether the learner answered exercise $j$ correctly, incorrectly, or did not practice it, respectively. It is important to note that $r_{u,j}$ includes information about unpracticed exercises, differing slightly from $y_{u,j}$ introduced in the background section. The similarity score between a pair of learners $u_i$ and $u_{i'}$ is then computed using their response vectors $\mathbf{r}_{u_i}$ and $\mathbf{r}_{u_{i'}}$, denoted as $sim(\mathbf{r}_{u_i}, \mathbf{r}_{u_{i'}})$. In our implementation, $sim(\cdot)$ is defined as Cosine Similarity due to its simplicity; however, other similarity functions, such as the inner product, could also be utilized.

After obtaining the similarity scores for each pair of learners within the batch, we can easily select the $p$ most similar learners as positive samples $\mathcal{U}_i^{b+}$ and the $q$ least similar learners as negative samples $\mathcal{U}_i^{b-}$ for each learner $u_i$ in the batch. Based on this, the contrastive learner-learner learning objective is defined as follows:

$$\mathcal{L}_{i,j}^{cll} = - \sum_{u_i^+ \in \mathcal{U}_i^{b+}} \left( box(u_i) \cap box(u_i^+) \right) + \sum_{u_i^- \in \mathcal{U}_i^{b-}} \left( box(u_i) \cap box(u_i^-) \right).$$
$$(10)$$

***Contrastive Learner-exercise Objective.*** For a learner $u_i$, the positive exercise samples consist of the exercises that $u_i$ has correctly answered in the training batch, while the negative exercise samples include those that $u_i$ has either answered incorrectly or has not practiced within the same batch. We denote the positive and negative exercise sets in the batch for learner $u_i$ as $\mathcal{E}_i^{b+}$ and $\mathcal{E}_i^{b-}$, respectively. Formally, the contrastive learner-exercise objective can be expressed as:

$$\mathcal{L}_{i,j}^{cle} = - \sum_{e_j^+ \in \mathcal{E}_i^{b+}} \left( box(u_i) \cap box(e_j^+) \right) + \sum_{e_j^- \in \mathcal{E}_i^{b-}} \left( box(u_i) \cap box(e_j^-) \right).$$
$$(11)$$

## 4.3 Gumbel-based Volume Objective

Directly optimizing box embeddings using the basic overlapping volume, i.e., Eq. (4), presents the second challenge of gradient vanishing, when two boxes do not intersect. This phenomenon hinders gradient-based training methods from effectively optimizing the model to meet the instinctive response prediction goals in CD, Eq. (8). Additionally, it complicates the process of ensuring that positive pairs overlap and negative pairs are separated in contrastive learning tasks, Eq. (10) and Eq. (11).

To address this issue, we draw inspiration from the work of Dasgupta et al. [6] and propose treating the standard box embeddings as Gumbel boxes. In this approach, we assume that the parameters of the box embeddings follow independent Gumbel distributions. Consequently, overlapping boxes, such as $box(u_i) \cap box(e_j)$, are generated from these Gumbel distributions based on their corresponding

vanilla box embeddings $box(u_i)$ and $box(e_j)$. This methodology ensures that all parameters remain active in gradient updates, even when the boxes are disjoint. Formally, the Gumbel distributions are defined as follows:

$$f(x; \mu, \beta) = \frac{1}{\beta} \exp\left(-\frac{x-\mu}{\beta} - \exp\left(-\frac{x-\mu}{\beta}\right)\right), \quad (12)$$

where $\beta$ controls the scale of the distribution, and $\mu$ governs the mean of the distribution. To avoid confusion, we denote the new lower and upper boundaries of overlapping boxes $box(u_i) \cap box(e_j)$ following Gumbel distributions as $\mu_{ij}^{\wedge}$ and $\mu_{ij}^{\vee}$. Each dimension $k$ of $\mu_{ij}^{\wedge}$ and $\mu_{ij}^{\vee}$ is calculated by

$$\mu_{ij,k}^{\wedge} := \min\left(u_{i,k}^{\vee}, e_{j,k}^{\vee}\right) \sim \text{Gumbel}\left(-\beta \ln\left(e^{-\frac{u_{i,k}^{\wedge}}{\beta}} + e^{-\frac{u_{j,k}^{\wedge}}{\beta}}\right), \beta\right),$$

$$\mu_{ij,k}^{\vee} := \max\left(u_{i,k}^{\wedge}, e_{j,k}^{\wedge}\right) \sim \text{Gumbel}\left(\beta \ln\left(e^{\frac{u_{i,k}^{\vee}}{\beta}} + e^{\frac{u_{j,k}^{\vee}}{\beta}}\right), \beta\right). \quad (13)$$

Next, the overlapping volume is calculated by the expected length for each dimension,

$$V(box(u_i) \cap box(e_j)) = \text{Sigmoid}\left(\mathbb{E}\left[\max\left(0, \mu_{ij,k}^{\vee} - \mu_{ij,k}^{\wedge}\right)\right]\right)$$

$$= \text{Sigmoid}\left(\prod_{k=1}^{d} \beta \log\left(1 + e^{-\left(\mu_{ij,k}^{\vee} - \mu_{ij,k}^{\wedge}\right)/\beta - 2\gamma}\right)\right), \quad (14)$$

where $\gamma$ is Euler-Mascheroni constant. The detailed derivation and proof are given in [6]. Equipped with Eq. (14) to calculate the overlapping volume, the above loss functions Eq. (9), Eq. (10) and Eq. (11) can be optimized across different training scenarios.

## 4.4 Model Training

To jointly learn the discriminative box embeddings for cognitive diagnosis, we integrate the response fitting task (Eq. (9)) with the additional contrastive box learning tasks (Eq. (10) and Eq. (11)) to obtain the final loss function:

$$\mathcal{L} = \sum_{\mathcal{R}^b \subset \mathcal{R}} \frac{1}{|\mathcal{R}^b|} \sum_{R_{i,j} \in \mathcal{R}^b} \left(\mathcal{L}_{i,j}^r + \alpha \left(\mathcal{L}_{i,j}^{cll} + \mathcal{L}_{i,j}^{cle}\right)\right). \quad (15)$$

where $\mathcal{R}^b \in \mathcal{R}$ denote a batch of response data and $\alpha$ is a coefficient to control the contrastive learning influence.

## 5 Response Inference & Cognitive State Output

After the training stages, we can obtain the optimized discriminative box embeddings for each learner $u_i \in \mathcal{U}$ and exercise $e_j \in \mathcal{E}$ through model inference. In this section, we will first demonstrate a highly efficient rank-based response prediction strategy (see § 5.1) to address the third technical challenge related to response prediction efficiency. Subsequently, we will introduce the process of obtaining numeric representations of learners' cognitive states (see § 5.2), which serves as a crucial foundation for further personalized applications in digital education [14, 32].

## 5.1 Efficient Response Prediction

To enhance the efficiency of response predictions for exercises that each learner has not yet practiced, we leverage the geometric properties of box embeddings. This approach streamlines the computation required to determine whether the boxes overlap, allowing us to efficiently assess whether each learner can correctly answer a given exercise. Since the probability of answering incorrectly for unpracticed exercises is zero, there is no need to predict the performance for such cases. Consequently, this narrows the scope of exercises for which the probability of answering correctly needs further prediction, thereby improving efficiency.

**Response Correctness Inference.** As mentioned above, box $box(u_i)$ of the learner $u_i$ and the box $box(e_j)$ of the exercise $e_j$ that $u_i$ cannot correctly answer are disjoint when there exists at least one dimension $k$ such that $\max(u_{i,k}^{\wedge}, e_{j,k}^{\wedge}) > \min(u_{i,k}^{\vee}, e_{j,k}^{\vee})$. This means the following two situations:

- The lower bound $u_{i,k}^{\wedge}$ of the learner box $box(u_i)$ is larger than the upper bound $e_{j,k}^{\vee}$ of the exercise box $box(e_j)$.
- The upper bound $u_{i,k}^{\vee}$ of the learner box $box(u_i)$ is smaller than the lower bound $e_{j,k}^{\wedge}$ of the exercise box $box(e_j)$.

Based on the above two cases, the key point in determining whether the learner box $box(u_i)$ and the exercise box $e_j$ overlap or are disjoint is to compare the size of their boundaries in each dimension $k$, i.e., $u_{i,k}^{\wedge}$ and $e_{j,k}^{\vee}$, or $u_{i,k}^{\vee}$ and $e_{j,k}^{\wedge}$, respectively. Therefore, we sort the lower and upper boundaries of each dimension $k$ for each exercise $e_j \in \mathcal{E}$. The ascending sorted box indices with respect to the lower and upper bound sets are denoted as $\left\{e_{k,1}^{\wedge}, e_{k,2}^{\wedge}, \ldots, e_{k,|\mathcal{E}|}^{\wedge}\right\}$ and $\left\{e_{k,1}^{\vee}, e_{k,2}^{\vee}, \ldots, e_{k,|\mathcal{E}|}^{\vee}\right\}$, respectively. Hereby, the sorted lower and upper bound sets, $[box(\mathcal{E})]_{i,k}^{\wedge}$ and $[box(\mathcal{E})]_{i,k}^{\vee}$, are given as:

$$[box(\mathcal{E})]_{i,k}^{\wedge} = \left\{\left(e_{k,1}^{\wedge}\right)_k^{\wedge}, \left(e_{k,2}^{\wedge}\right)_k^{\wedge}, \ldots, \left(e_{k,|\mathcal{E}|}^{\wedge}\right)_k^{\wedge}\right\},$$

$$[box(\mathcal{E})]_{i,k}^{\vee} = \left\{\left(e_{k,1}^{\vee}\right)_k^{\vee}, \left(e_{k,2}^{\vee}\right)_k^{\vee}, \ldots, \left(e_{k,|\mathcal{E}|}^{\vee}\right)_k^{\vee}\right\}. \quad (16)$$

For each dimension $k$, the lower bound $u_k^{\wedge}$ and upper bound $u_k^{\vee}$ of the learner box serve as keys for searching within the sorted upper bounds $[box(\mathcal{E})]_{i,k}^{\wedge}$ and lower bounds $[box(\mathcal{E})]_{i,k}^{\vee}$ of the exercise sets $\mathcal{E}$, respectively. The search operation identifies two position indices $e_i^+$ and $e_i^-$, ensuring that $\left(e_{k,e_i^+}^{\wedge}\right)_k^{\wedge} < u_{i,k}^{\vee} \leq \left(e_{k,e_i^++1}^{\wedge}\right)_k^{\wedge}$ and $\left(e_{k,e_i^--1}^{\vee}\right)_k^{\vee} \leq u_{i,k}^{\wedge} < \left(e_{k,e_i^-}^{\vee}\right)_k^{\vee}$. Exercises indexed between $e_{i,k}^+$ and $e_{i,k}^-$ indicate that learners can correctly respond, denoted as $\mathcal{E}_i^+$, while those before $e_{i,k}^+$ and after $e_{i,k}^-$ are exercises they cannot solve correctly, denoted as $\mathcal{E}_i^-$.

**Correct Response Probability Calculation.** After obtaining the exercise sets $\mathcal{E}_i^+$ and $\mathcal{E}_i^-$ for each learner, we can first ensure that the probability that $u_i$ correctly answers each exercise in $\mathcal{E}_i^-$ is always 0 since the learner $u_i$ cannot correctly solve them. This step narrows down the cost of calculated probabilities. Then, we can infer the probability that each learner $u_i$ correctly answers each exercise $e_j \in \mathcal{E}_i^+$ by input the learner box embedding $box(u_i)$ and each exercise box embedding $box(e_j)$, $e_j \in \mathcal{E}_i^+$ to our model

and calculate the probability $\hat{y}_{i,j} = V(box(u_i) \cap box(e_j))$ based on Eq. (14).

**Time Complexity Analysis.** The set of unpracticed exercises for each learner $u_i \in \mathcal{U}$ is denoted as $\mathcal{E}_i := \mathcal{E}_i^+ \cup \mathcal{E}_i^-$, with an average probability $p_i = |\mathcal{E}_i^+|/|\mathcal{E}_i|$ of answering correctly. This indicates that the box representing learner $u_i$ has a probability $p_i$ of intersecting with each exercise's box, while the average probability of disjointness is $(1 - p_i)$.

Next, we discuss the time complexity. The total time cost consists of two components:

- *Infer response correctness.* By utilizing a classical sorting algorithm (e.g., Quick Sort [12]), the time complexity for sorting the $d$-dimensional box embeddings of $\mathbb{E}_{u_i \sim \mathcal{U}}|\mathcal{E}_i|$ unpracticed exercises can be expressed as: $O\left(\mathbb{E}_{u_i \sim \mathcal{U}} d \cdot |\mathcal{E}_i| \log \mathbb{E}_{u_i \sim \mathcal{U}}|\mathcal{E}_i|\right)$.
- *Infer response probability.* The time cost for this operation is determined by the expected size of $\mathbb{E}_{u_i \sim \mathcal{U}}|\mathcal{E}_i^+|$, resulting in a time complexity of: $O\left(\mathbb{E}_{u_i \sim \mathcal{U}} \left(d \cdot |\mathcal{E}_i^+|\right)\right) \Leftrightarrow O\left(\mathbb{E}_{u_i \sim \mathcal{U}} \left(d \cdot |\mathcal{E}_i| \cdot p_i\right)\right)$.

## 5.2 Learner Cognitive State Output

Traditional vector embedding-based CD models typically diagnose either the latent problem-solving ability or the mastery probability of specific knowledge concepts. In contrast, BoxCD utilizes the flexibility of box operations to simultaneously represent both aspects of cognitive states, thereby offering a more comprehensive learner model. We present a **Case Study** in Appendix C to illustrate the diagnostic output generated by BoxCD.

**Problem-solving Ability.** Problem-solving ability positively correlates with the probability of answering correctly, which can be represented by the intersection volume of the learner's and exercise boxes. To encapsulate each learner $u_i$'s problem-solving ability, we define their box embedding $box(u_i^a)$ based on overall response performance. To achieve this, we introduce a novel box accumulation operation defined as follows:

---

**Box Accumulation**

The accumulation of multiple boxes, specially customized for BoxCD, refers to the process of summing a set of boxes with any overlap among them is considered only once. Given $n$ box representations $box(x_1), box(x_2), \ldots, box(x_n)$, the accumulated box is denoted as $\bigoplus_{i=1}^{n} box(x_i)$, of which volume is calculated by summing the individual volumes of each box with any overlapping volume counted once:

$$V\left(\bigoplus_{i=1}^{n} box(x_i)\right) = \sum_{i=1}^{n} \prod_{k=1}^{d} \left(x_{i,k}^{\vee} - x_{i,k}^{\wedge}\right) - \sum_{i=1}^{n} \sum_{j=n+1}^{n} V(box(x_i) \cap box(x_j)) \in \mathbb{R}^1. \quad (17)$$

---

The learner ability is reflected by the accumulation of all the intersections between $u_i$'s original box embeddings $box(u_i)$ and

each exercise box representation $box(e_j)$ with $j = 1, 2, \ldots, M$.

$$box(u_i^a) = \bigoplus_{j=1}^{M} \left(box(u_i) \cap box(e_j)\right). \quad (18)$$

We calculate the single-aspect ability using the accumulation volume operation: $\mathbf{u}_i^a = Sigmoid\left(V\left(box(u_i^a)\right)\right) \in \mathbb{R}^1$. This approach aligns with traditional CD models with $d = 1$, such as IRT. To capture multi-aspect abilities, akin to MIRT and MCD, we need to define a box flatten operation that transforms the accumulation box into a $d$-interval tie, resulting in a $d$-dimensional vector, as follows:

---

**Flatten of Multiple Boxes**

Flattening multiple boxes involves projecting the multiple box embeddings into a flat vector space. Given a set of box representations $box(x_1), box(x_2), \ldots, box(x_n)$, flattening over them can be achieved by:

$$\asymp (box(x_1), box(x_2), \ldots, box(x_n))$$
$$= \left\langle \left[\| a_{1,1}, a_{2,1}, \ldots, a_{n,1} \|\right], \left[\| a_{1,2}, a_{2,2}, \ldots, a_{n,2} \|\right], \right.$$
$$\left. \ldots, \left[\| a_{1,d}, a_{2,d}, \ldots, a_{n,d} \|\right] \right\rangle \in \mathbb{R}^d, \quad (19)$$

where the $k$-dimensional of vector $\asymp (box(x_k), box(x_k), \ldots, box(x_k))$ is the cumulative length of $n$ intervals without repeatedly considering overlappng regions,

$$\| a_{1,k}, a_{2,k}, \ldots, a_{n,k} \| = \sum_{i=1}^{n} \left(x_{i,k}^{\vee} - x_{i,k}^{\wedge}\right)$$
$$- \sum_{i=1}^{n} \sum_{j=n+1}^{n} \max\left(0, \min\left(x_{i,k}^{\vee}, x_{j,k}^{\vee}\right) - \max\left(x_{i,k}^{\wedge}, x_{j,k}^{\wedge}\right)\right) \in \mathbb{R}^1. \quad (20)$$

---

Then, we have $\mathbf{u}_i^a = Sigmoid\left(\asymp \left(box(u_i^a)\right)\right) \in \mathbb{R}^d$, where each element $u_{i,k}^a$ denotes an ability factor in one of the $d$ aspects.

**Knowledge Mastery Probability.** BoxCD computes learners' mastery probability of specific knowledge concepts in two steps. First, it obtains knowledge concept embeddings; second, it calculates knowledge mastery by fusing knowledge and learner embeddings, following a similar pipeline proposed in [8, 33]. Specifically, BoxCD represents the box embedding of each knowledge concept $c_k$ as the union of all exercise boxes that assess $c_k$:

$$box(c_k) = box(x_1) \cup box(x_2) \cup \ldots \cup box(x_n) \quad (21)$$

which is a popular operation for representing knowledge concepts in vector embedding-based CD models [33]. Subsequently, the mastery box embedding concerning knowledge concept $c_k$ is determined by the intersection of the learner's box $box(u_i)$ and the box representing the knowledge concept $box(c_k)$. The scalar mastery probability is represented as the box volume, $V(box(u_i) \cap box(c_k))$, which corresponds to $u_{i,k}$ in traditional CD models, such as NCDM.

| Dataset | Model | ACC ↑ | AUC ↑ | F1-score ↑ | RMSE ↓ |
|---------|-------|-------|-------|-----------|--------|
| ASSIST | IRT | 65.63 | 70.90 | 79.25 | 47.31 |
|  | MIRT | 65.64 | 68.61 | 79.25 | 48.95 |
|  | MCD | 67.26 | 73.47 | 79.94 | 45.38 |
|  | NCDM | 73.63 | 76.73 | 80.25 | 42.64 |
|  | KaNCD | 73.05 | 76.58 | 81.60 | 42.67 |
|  | RCD | 72.81 | 76.75 | 80.54 | 42.39 |
|  | DCD | 61.49 | 62.77 | 70.91 | 47.34 |
|  | ID-CDM | 73.16 | 76.54 | 80.83 | 42.76 |
|  | BoxCD | **73.87** | **77.25** | **82.31** | **42.23** |
| Junyi | IRT | 68.21 | 78.35 | 80.10 | 43.46 |
|  | MIRT | 72.20 | 78.33 | 80.97 | 42.48 |
|  | MCD | 73.04 | 79.90 | 81.46 | 41.91 |
|  | NCDM | 72.86 | 78.06 | 80.75 | 42.40 |
|  | KaNCD | 76.14 | 81.18 | 82.87 | 40.45 |
|  | RCD | 76.95 | 82.29 | 83.20 | 39.84 |
|  | DCD | 76.41 | 78.01 | 80.48 | 42.19 |
|  | ID-CDM | 65.95 | 68.82 | 69.97 | 53.06 |
|  | BoxCD | **77.38** | **82.83** | **83.69** | **39.21** |

**Table 1: Performance comparison. The best performance is highlighted in bold. ↑ (↓) means the higher (lower) score the better (worse) performance, the same as below.**

## 6 Experiments

### 6.1 Experimental Settings

*Datasets*. We evaluate BoxCD and the baseline models on two representative datasets: ASSIST [7] and Junyi [3]. The statistics for these datasets are provided in Table 4, with more detailed descriptions in Appendix A.

*Baselines*. The baselines include typical latent factor models from educational psychology, such as IRT [10], MIRT [1], the matrix factorization-based MCD [24], and deep learning models like NCDM [33], RCD [8], KaNCD [34], ID-CDM [16], and DCD [39]. More details about the baselines are provided in Appendix B.

*Evaluation*. Since cognitive states are not directly observable, CDMs are generally evaluated through student performance prediction tasks on test datasets [2]. To evaluate prediction performance, we use ACC, AUC, and F1-score as metrics for binary classification (thresholded at 0.5) based on whether the response is correct. Additionally, we apply RMSE as a regression metric for correct response probability, following previous work [8].

*Implementation*. We split all datasets into training, validation, and test sets using a 7:1:2 ratio. For IRT, the dimension size $d$ is set to 1, while for other models, $d$ corresponds to the number of knowledge concepts. The mini-batch size is 256. During training, we select the learning rate $lr$ from $\{0.001, 0.002, 0.005, 0.01\}$, with $p, q \sim \{3, 5, 10\}$, $\alpha \sim \{0.1, 0.5, 1, 5, 10\}$, $\beta = 1$, and $\gamma = 1$. The optimal setups are $lr = 0.002, p = 5, q = 5, \alpha = 1$ for AS-SIST, and $lr = 0.002, p = 5, q = 10, \alpha = 1$ for Junyi. All network parameters are initialized using Xavier initialization [9]. Each model is implemented in PyTorch [25] and optimized with the Adam optimizer [15]. Each experiment is repeated five times, and the average scores are reported. All experiments are conducted on a Linux server equipped with two 3.00GHz Intel Xeon Gold 5317 CPUs and one Tesla A100 GPU. Our code is available at https://anonymous.4open.science/r/BoxCD.

| Model | Latency (s) ↓ on Assist | | Latency (s) ↓ on Junyi | |
|-------|:-----------:|:-----------:|:-----------:|:-----------:|
|  | Correctness | Probability | Correctness | Probability |
| IRT | 8.21 | 6.14 | 14.63 | 8.52 |
| BoxCD ($d$=1) | **4.62** | **5.07** | **7.73** | **8.38** |
| MIRT | 16.86 | 12.11 | 24.93 | 17.79 |
| MCD | 11.82 | 7.67 | 17.22 | 11.29 |
| NCDM | 10.58 | 8.42 | 20.50 | 11.57 |
| KaNCD | 27.03 | 17.93 | 40.13 | 24.61 |
| RCD | 34.52 | 24.06 | 303.52 | 244.26 |
| DCD | 18.81 | 11.38 | 20.59 | 16.31 |
| ID-CDM | 12.85 | 12.82 | 22.79 | 22.76 |
| BoxCD | **4.03** | **6.13** | **9.34** | **11.22** |

**Table 2: The latency time on predicting the response correctness and the correct response probability on test data.**

### 6.2 Prediction Results and Analysis

*Effectiveness*. Table 1 presents the prediction performance of BoxCD compared to baseline models in the learner response prediction task. BoxCD consistently exceeds the performance of baseline models across all datasets. These gains primarily result from modeling both learners and exercises as boxes in the latent space. Baseline models use fixed vector representations for CD modeling, which do not accommodate fluctuations in learner states and exercise semantic uncertainty.

*Efficiency*. We compare the inference efficiency of each model by measuring the prediction time on test sets. Table 2 displays the inference time for predicting both binary response correctness and correct response probability. Specifically, we include the BoxCD with the $d = 1$ setting to compare it with IRT ($d = 1$). The following observations can be made: (1) Compared to vectorized response prediction (i.e., all the baselines), we achieve better average inference latency. This improvement arises because we can filter out a large proportion of incorrect response predictions using fast rank-based operations, demonstrating the efficiency of the proposed box embedding-based operation. (2) Baseline models predict probabilities faster than they determine correctness since current vector-based CD models first infer the probability of correctness and then classify responses based on a threshold. In contrast, BoxCD operates differently: it first classifies the correctness and only calculates the probability for items corresponding to correct responses. Consequently, BoxCD's correctness classification is faster than the probability computation.

*Ablation Study*. We investigate the effects of each key component of BoxCD. The results in Figure 2 illustrate the performance of BoxCD under various conditions: the basic BoxCD defined in § 4.1 (denoted as vanilla), removal of the contrastive learner-learner loss (w/o $\mathcal{L}^{cll}$), removal of the contrastive learner-exercise loss (w/o $\mathcal{L}^{cle}$), and removal of the Gumbel-based volume objective (w/o Gumbel) across two datasets. The results reveal the following: (1) Removing any component negatively impacts BoxCD's performance. (2) Incorporating either contrastive loss, when paired with Gumbel-based optimization, enhances the accuracy of the vanilla model. However, the effect of the contrastive loss diminishes when the Gumbel mechanism is removed, indicating that Gumbel-based

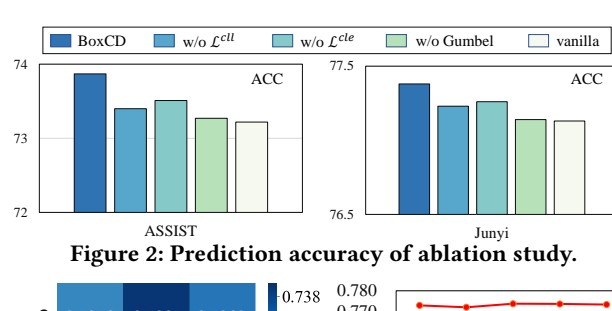

Figure 2: Prediction accuracy of ablation study.

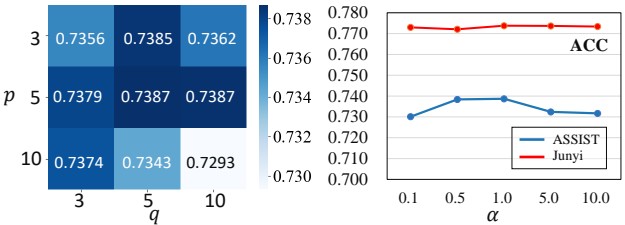

Figure 3: ACC scores of BoxCD with: (Left) varying sampling numbers, and (Right) different $\alpha$ values.

optimization is crucial for mitigating gradient vanishing in the context of contrastive learning.

## 6.3 Parameters Sensitivity

**Impact of Sampling Number.** Figure 3 (left part) illustrates the impact of the number of positive ($p$) and negative ($q$) sample selections in the learner's contrastive loss on ASSIST data. As shown in the figure, with the increase in either $p$ or $q$, the model performance begins to rise, indicating that introducing contrastive learning among learners enhances the model. However, once a certain threshold is reached, the performance stabilizes, suggesting that the gains from contrastive learning are limited.

**Impact of $\alpha$.** Figure 3 illustrates the impact of the parameter $\alpha$ in the final loss function (Eq. (15)) on model performance. We observe that the model performs optimally when $\alpha$ is around 1. Both excessively small and large values of $\alpha$ result in a decline in prediction performance.

## 6.4 Box Representation Analysis

**Uncertainty.** We compare the uncertainty captured by BoxCD (e.g., fluctuations in learner states) with statistical uncertainty from the data to evaluate the rationality of the box representation. The interval length in each dimension reflects uncertainty: longer intervals indicate higher uncertainty, while shorter intervals suggest lower uncertainty. More response records for a learner or task result in more accurate modeling and reduced uncertainty [35]. Table 3 presents the mean interval lengths of box representations for all learners and exercises, normalized to the 0-1 range using min-max scaling [11]. It also shows the average number of exercises attempted by each learner and the average number of learners per exercise. The Junyi dataset has more learners per exercise but fewer exercises per learner compared to ASSIST, indicating lower uncertainty in exercise boxes but higher uncertainty in learner boxes. Consequently, the mean interval lengths for learner boxes in Junyi are higher, while those for exercise boxes are lower, validating the effectiveness of BoxCD's uncertainty modeling.

| Statistic | ASSIST | Junyi |
|---|---|---|
| Interval mean of learner boxes | 0.4217 | 0.5322 |
| Interval mean of exercise boxes | 0.7124 | 0.6923 |
| Interacted exercise number per learner | 66.99 | 39.34 |
| Interacted learner number per exercise | 15.71 | 109.73 |

Table 3: Statistic results for uncertainty analysis.

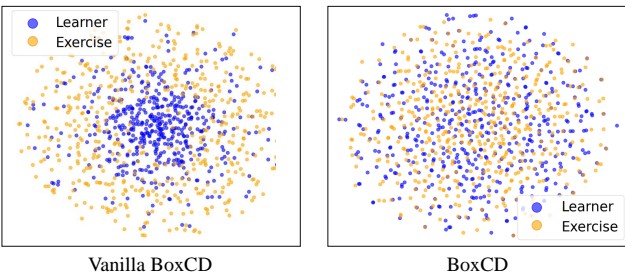

Figure 4: The visualization of learner and exercise boxes.

**Visualization.** To investigate the contribution of contrastive learning loss to box modeling, we visualize the learner and exercise boxes generated by both BoxCD and the vanilla BoxCD (i.e., the basic version from § 4.1, which does not incorporate contrastive learning). For visualization, we transform both the learner and exercise boxes into vectors using the flatten operation. Figure 4 demonstrates that the vanilla model, lacking box contrastive learning, results in data points clustering together, particularly among learner points. In contrast, BoxCD effectively prevents the aggregation of each box, leading to a more uniform distribution. This highlights the importance of differentiation between learners and exercises in education [5].

## 7 Conclusion

This work focuses on assessing learners' cognitive states in the educational context through Cognitive Diagnosis (CD). It highlights the challenges of existing CD methods regarding effectiveness and efficiency. These challenges stem from their reliance on vectorized representations, which fail to capture the diversity and uncertainty of learners and exercises. Additionally, the time-consuming nature of response predictions exacerbates these issues. To address these challenges, we propose a contrastive probabilistic Box embedding model for Cognitive Diagnosis (BoxCD). This model employs probabilistic box embeddings to represent learners and exercises more accurately in CD tasks. We also introduce contrastive learning objectives to enhance the stability of the box embeddings. Finally, we present a rank-based response prediction method that leverages box intersections for faster predictions. Experimental results demonstrate that BoxCD significantly outperforms existing models, underscoring its potential to enhance personalized learning experiences on digital education platforms. As educational technologies continue to evolve, BoxCD represents a vital advancement in harnessing cognitive diagnosis to better support learner success.

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

# A Dataset

| Statistic | ASSIST | Junyi |
|---|---|---|
| Number of learners | 4,163 | 1,000 |
| Number of questions | 17,746 | 835 |
| Number of knowledge concepts | 123 | 835 |
| Number of concepts per exercise | 1.21 | 1 |
| Number of response records | 267,416 | 353,835 |
| #correct records / #incorrect records | 65.77% | 65.17% |

**Table 4: The statistics of two datasets.**

We conduct experiments on two real-world datasets: ASSIST [7] and Junyi [3]. The statistics of these datasets are summarized in

Table 4. For all datasets, we retain the first-time exercise-answering records for the same learner-exercise pairs to facilitate cognitive diagnosis, aligning with common practices in previous studies [33]. Detailed information on the datasets and preprocessing methods is provided below:

- **ASSIST (ASSISTments 2009-2010 "skill builder")** [7] This dataset is an open resource collected by the ASSISTments online tutoring system[1], which has become a popular benchmark for cognitive diagnosis. We retain learners with more than 15 response records in ASSIST to ensure that each learner has sufficient data for diagnosis. Additionally, since ASSIST does not provide the knowledge concept graph required by the baseline RCD [8], we employ a statistical method proposed in RCD to automatically generate the knowledge concept graph.

- **Junyi** [3] This dataset comprises online learning logs collected from Junyi Academy, a Chinese online educational platform[2]. It explicitly provides knowledge concept graphs, which support the baseline model (i.e., RCD [8]) that requires knowledge concept connections. Junyi is increasingly used for evaluating online education tasks [5, 8]. We randomly select 1,000 learners with more than 15 practice records to ensure sufficient data for diagnosis.

## B  Baseline

The baselines include the typical latent factor models derived from educational psychology, i.e., IRT [10], MIRT [1], the Matrix Factorization-based MCD [24], and the deep learning-based models NCDM [33], RCD [8], KaNCD [34], ID-CDM [16] and DCD [39].

- IRT [10]: IRT models unidimensional learners and exercises' features with a logistic-like function.
- MIRT [1] extends the representation of learners and exercises in IRT from one-dimensional to multidimensional.
- MCD [24] predicts learner performance by factoring score matrix and get learners and exercises' latent vectors.
- NCDM [33] is one of the most popular deep learning-based CD methods, which models high-order and complex student-exercise interaction functions with MLPs.
- KaNCD [34] extends NCDM by extending NeuralCD with the knowledge associations consideration into NCDM to improve the diagnostic results.
- RCD [8] is the first KCG-based cognitive diagnosis model, introducing relations between knowledge concepts and modeling these relations using a graph structure.
- ID-CDM [16] extends the previous CD methods to extract the initial features of learners and exercises from response data.
- DCD [39] disentangles learner representations to learn discriminative learner cognitive states.

## C  Case Study

We present the cognitive state diagnosis results obtained using BoxCD. Specifically, we randomly selected a learner (ID=250) from the Junyi dataset, whose overall correct rate is 0.7713. Figure 5

| Knowledge Concept | Correct Rate |
|---|---|
| Algebra | 0.77 |
| Function | 0.72 |
| Advanced Vector | 0.68 |
| Derivative | 0.57 |
| Basic Trigonometry | 0.55 |
| Number | 0.43 |

**Table 5: Response statistics of a learner.**

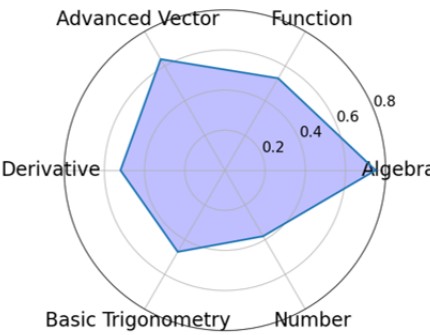

**Figure 5: The visualization of the learner proficiency on several knowledge concepts learned by BoxCD.**

shows the cognitive state learned by BoxCD based on § 5.2, including the learner's overall ability (0.6732) and the mastery levels across six knowledge concepts. Additionally, we calculate the correct response rates on exercises related to each knowledge concept based on the learner's original response data, summarized in Table 5. The diagnosed ability aligns with the learner's overall correct rate, and the mastery levels of knowledge concepts positively correlate with their accuracy on the corresponding exercises, adhering to the psychological monotonicity assumption [33]. This correlation reflects the rationality of the BoxCD diagnostic output. These numerical representations of learners' cognitive states serve as a crucial foundation for further personalized applications in digital education [13, 32].

---

[1]https://sites.google.com/site/assistmentsdata/

[2]https://www.junyiacademy.org/

