# OpenReview forum: "BoxCD: Leveraging Contrastive Probabilistic Box Embedding for Effective and Efficient Learner Modeling"
_ACM.org/TheWebConf/2025/Conference — WWW 2025 Poster_

### Official Review · Reviewer_USh2 · 2024-11-27

**Novelty:** 4
**Technical Quality:** 4

**Review:**

This paper uses box embeddings combined with contrastive learning and a Gumbel-based volume objective for cognitive diagnosis. The proposed method introduces a geometric perspective by embedding both learners and exercises in box spaces and enhances traditional CDMs by focusing on monotonicity and semantic constraints within the embedding space. Experiments conducted on multiple educational datasets show consistent performance improvements over baseline models such as RCD and NCDM.

Strengths:
1. The use of box embeddings provides an intuitive geometric interpretation of learner and exercise relationships, aligning well with psychological principles like monotonicity in cognitive diagnosis

2. The rank-based response prediction mechanism demonstrates improved computational efficiency compared to traditional vector-based models, which is a notable advancement for large-scale online learning platforms.

Weaknesses:

1. The motivation is somewhat incremental, focusing on the embedding-based cognitive diagnosis, which has already been explored using vectors and probabilistic methods. The novelty of transitioning to box embeddings does not significantly push the boundaries of the field beyond established paradigms

2. While the paper introduces a contrastive learning approach and Gumbel-based volume objectives, these techniques are adaptations of existing methods applied in other domains like knowledge graphs.

3. The performance improvements reported, while consistent, are marginal in some cases compared to existing baselines like RCD and NCDM

**Questions:**

N/A

**Reviewer Confidence:**

3: The reviewer is confident but not certain that the evaluation is correct

**Scope:**

3: The work is somewhat relevant to the Web and to the track, and is of narrow interest to a sub-community

---

### Official Review · Reviewer_6w8F · 2024-11-27

**Novelty:** 4
**Technical Quality:** 4

**Review:**

## Summary

This paper presents BoxCD, a novel cognitive diagnosis (CD) model for digital education platforms. The model leverages contrastive probabilistic box embeddings to represent learners and exercises as high-dimensional axis-aligned hyper-rectangles (boxes). By utilizing the volume of intersecting boxes, BoxCD predicts learners' responses, effectively capturing semantic diversity and uncertainty in cognitive states.

To enhance the stability of box embeddings and improve diagnostic effectiveness, the authors integrate contrastive learning objectives with response prediction goals. They also develop a rank-based response prediction method that exploits the geometric properties of box embeddings to efficiently assess learners' response correctness. Experiments on two real-world datasets demonstrate that BoxCD outperforms traditional CD models in both effectiveness and efficiency.

---

## Evaluation

### **Quality**

- The paper is technically sound and introduces a well-motivated approach to address the limitations of existing CD models. The use of probabilistic box embeddings to capture the diversity and uncertainty of learners' cognitive states is innovative. The mathematical formulations are presented clearly, and the integration of contrastive learning objectives is justified.

- However, the experimental analysis could be more comprehensive. The evaluation is limited to two datasets, which may not fully demonstrate the model's generalizability. Additionally, the paper could provide more in-depth explanations of the baseline models and how they compare to BoxCD.

---

### **Clarity**

- The paper is generally well-written and organized, making it accessible to readers with a background in cognitive diagnosis and embedding techniques. The introduction effectively sets the context and motivation for the work.

    - Mathematical Notations: The notation in Section 3.3 on probabilistic box embeddings is dense. Providing illustrative examples or a detailed workflow diagram for the model training could enhance understanding.
    - Baseline Models: A more thorough explanation of the baseline models would aid in understanding the comparative results.

---

### **Originality**

- While applying probabilistic box embeddings to cognitive diagnosis tasks introduces a new perspective, the use of contrastive learning to enhance embedding stability is not novel and has been widely explored in existing literature. Contrastive learning is a well-established technique for improving representation learning and has been applied in various domains.

- Consequently, the integration of contrastive probabilistic box embeddings in this work may be seen as an incremental advancement rather than a significant innovation. The paper could strengthen its contribution by providing a deeper analysis of how this specific combination uniquely addresses challenges in CD tasks and differentiates from prior methods that also employ contrastive learning for embedding stability.

---

### **Significance**

- BoxCD has the potential to significantly impact personalized learning in digital education platforms. By effectively capturing the diversity and uncertainty of learners' cognitive states and improving computational efficiency, the model addresses key challenges in the field. The proposed approach could enhance adaptive learning systems and provide more accurate diagnostics.

- **Limitations**:
    - The limited experimental scope may affect the perceived significance. Evaluating the model on additional datasets with varied characteristics (e.g., sparsity levels or domain-specific data) would strengthen the generalizability of the results.
    - Providing practical examples or case studies illustrating how the box embeddings improve interpretability would further enhance the contribution.

**Questions:**

### **Regarding the Model Training Workflow**
1. *Could you provide a detailed workflow diagram for the model training process?*
   - The diagram should illustrate how different components of the BoxCD model interact during training, helping readers better understand the overall process.

---

### **Regarding the Expansion of Experimental Analysis**
2. *Could the experimental analysis section be made more detailed?*
   - The current analysis is relatively brief. Consider expanding this section to provide more contextual support for the results.
   - *Explain the core characteristics and working principles of baseline models (e.g., IRT, MIRT, KaNCD, etc.).* This would help readers better understand the background and significance of the comparative results.

---

### **Regarding the Role of the Contrastive Learning Objective**
3. *Could you provide more detailed experimental evidence to demonstrate the impact of the contrastive learning objective on model performance?*
   - For instance, include more comprehensive quantitative analyses showing how removing or modifying the contrastive loss affects model performance and the stability of box embeddings.

4. *Could you expand the ablation study section?*
   - While the current results demonstrate the importance of various components, the details remain insufficient.

**Reviewer Confidence:**

2: The reviewer is willing to defend the evaluation, but it is likely that the reviewer did not understand parts of the paper

**Scope:**

4: The work is relevant to the Web and to the track, and is of broad interest to the community

---

### Official Review · Reviewer_xFng · 2024-11-29

**Novelty:** 5
**Technical Quality:** 5

**Review:**

This paper presents BoxCD, a contrastive probabilistic model for cognitive diagnosis that uses high-dimensional boxes to represent learners and exercises. By optimizing box embeddings through contrastive learning and a rank-based prediction method, it captures cognitive diversity and improves response prediction efficiency. Experiments show BoxCD outperforms traditional models in both effectiveness and efficiency for personalized learning.

Pros:

1. Cognitive diagnosis is an important research problem for education.

2. The reported experimental results look promising.

3. This paper is easy to read.


Cons:

1. I wonder whether the proposed method is the most simple and effective way to handle the so-called diversity problem in cognitive diagnosis. Based on the examples introduced in the Introduction section, it seems this problem can be easily handled by incorporating the environment factors into features.

2. Since I am not an expert in education and Probabilistic Box, I cannot understand why Probabilistic Box can handle the problem claimed by the authors intuitively.

3. I am not sure whether the baseline methods compared in experiments are SOTA.

**Questions:**

Please refer to above comments

**Reviewer Confidence:**

1: The reviewer's evaluation is an educated guess

**Scope:**

3: The work is somewhat relevant to the Web and to the track, and is of narrow interest to a sub-community

---

### Official Review · Reviewer_A6Sa · 2024-11-29

**Novelty:** 4
**Technical Quality:** 4

**Review:**

This paper focuses on the effectiveness and efficiency problems in cognitive diagnosis. It propose BoxCD, a contrastive probabilistic box embedding model for cognitive diagnosis. Its core idea is to utilize high-dimensional axis-aligned hyper-rectangles (boxes) to represent learners and exercises, with the volume of intersecting boxes used to predict learners' responses. Experiments on two public datasets demonstrate the effectiveness of BoxCD.

**Strengths**

- This paper is the first to utilize probabilistic box embedding techniques to represent learners and exercises.

- This paper focuses on the important problems in cognitive diagnosis which have been overlooked, namely, efficiency, and the solution, employing the volume as the prediction results of learners' responses seems intuitive and reasonable.

- The efficiency experiment appears to fully validate that the authors' method achieves faster prediction compared to existing approaches.

**Questions:**

**Weaknesses**

- **Novelty**. This paper appears to be inspired by works in retrieval, such as [1][2], as many of the formulas seem quite similar. Could the authors elaborate on the differences between their approach and these retrieval-based methods? Highlighting the distinctions would help clarify the unique contributions of this work.


- **Dinstinction and Advantage with UCD**. "Regarding effectiveness, current vectorized representations of learners and exercises inadequately capture their diversity and uncertainty. For instance, a learner's cognitive state and an exercise's features fluctuate within specific ranges depending on the context." The diversity and uncertainty in vectorized representations within cognitive diagnosis has been explored in UCD [3]. The authors should include [3] in the introduction and further clarify how their approach distinguishes itself from [3] and the advantages compared with [3].

- **Baselines**. In lines 741-742, the authors state, "while for other models, \(d\) corresponds to the number of knowledge concepts." However, for CDMs such as MIRT and KaNCD, fewer dimensions (e.g., 16 or 20) typically perform much better than using a dimensionality equal to the number of knowledge concepts. Specifically, in Table 1, KaNCD underperforms NCDM on the ASSIST dataset, which may conflict with findings in recently published papers [3][4][5]. This raises concerns about the fairness of the comparison. Furthermore, the latest reasonable baselines from 2024 conferences or journals should be considered [3][5][6][7][8], as ID-CDM is the only CDM in 2024 but falls in inductive paradigm. It performs extremely poorly in Table 1 on the Junyi dataset.


- **Mastery Level Acquisition**: The authors provide a link to the code repository. However, in `model.py`, the `get_knowledge_status` function appears to have issues. The code does not include `self.student_emb`, and the comments indicate it pertains to a `NeuralCDM`, which raises questions about its implementation. Furthermore, why do the authors not compare DOA, a widely used metric for evaluating the interpretability of CDMs? Interpretability is a critical aspect in cognitive diagnosis [3][4][5], and incorporating DOA would provide a more comprehensive evaluation of the model's performance.


[1] Liang, Tingting, et al. "Contrastive Box Embedding for Collaborative Reasoning." Proceedings of the 46th International ACM SIGIR Conference on Research and Development in Information Retrieval. 2023.

[2] Mei, Lang, et al. "Learning probabilistic box embeddings for effective and efficient ranking." Proceedings of the ACM Web Conference 2022.

[3] Wang, Fei, et al. "Unified Uncertainty Estimation for Cognitive Diagnosis Models." Proceedings of the ACM on Web Conference 2024. 2024.

[4] Wang, Fei, et al. "NeuralCD: a general framework for cognitive diagnosis." IEEE Transactions on Knowledge and Data Engineering 35.8 (2022): 8312-8327.

[5] Shen, Junhao, et al. "Symbolic Cognitive Diagnosis via Hybrid Optimization for Intelligent Education Systems." Proceedings of the AAAI Conference on Artificial Intelligence. Vol. 38. No. 13. 2024.

[6] Yao, Fangzhou, et al. "Adard: An adaptive response denoising framework for robust learner modeling." Proceedings of the 30th ACM SIGKDD Conference on Knowledge Discovery and Data Mining. 2024.

[7] Qian, Hong, et al. "ORCDF: An Oversmoothing-Resistant Cognitive Diagnosis Framework for Student Learning in Online Education Systems." Proceedings of the 30th ACM SIGKDD Conference on Knowledge Discovery and Data Mining. 2024.

[8] Gao, Weibo, et al. "Zero-1-to-3: Domain-Level Zero-Shot Cognitive Diagnosis via One Batch of Early-Bird Students towards Three Diagnostic Objectives." Proceedings of the AAAI Conference on Artificial Intelligence. Vol. 38. No. 8. 2024.

**Reviewer Confidence:**

4: The reviewer is certain that the evaluation is correct and very familiar with the relevant literature

**Scope:**

4: The work is relevant to the Web and to the track, and is of broad interest to the community

---

### Official Review · Reviewer_8Uuf · 2024-12-02

**Novelty:** 5
**Technical Quality:** 4

**Review:**

The authors propose a contrastive probabilistic box embedding model for cognitive diagnosis (BoxCD) in digital education, aiming to enhance the effectiveness and efficiency of learner modeling. BoxCD represents learners and exercises as high-dimensional axis-aligned hyper-rectangles (boxes). It predicts learners' responses by calculating the volume of intersecting boxes, effectively modeling the complexity of cognitive states. BoxCD integrates contrastive learning objectives to stabilize box embeddings, treating the standard box embeddings as Gumbel boxes and develops a rank-based response prediction method to improve prediction speed. Experimental results on two real-world datasets demonstrate that BoxCD outperforms traditional models in both accuracy and efficiency, showing its potential to enhance personalized learning experiences.

Novelty: The authors apply the existing probabilistic box embedding method to the cognitive diagnosis task and propose different solutions to address the technical challenges of integrating probabilistic box embeddings into cognitive diagnosis models. The proposed approach achieves good results; however, all the methods used are based on existing methods, making the overall novelty of the paper moderate.
## Props:
The overall structure of the paper is relatively clear, and the analysis of the experimental results is comprehensive.
## Cons:
1. The authors claim that the proposed method can address the inability of traditional CD methods to sufficiently capture diversity and uncertainty. However, based on the descriptions and experiments in the paper, it does not convincingly demonstrate that the aforementioned issues exist in traditional methods. It is recommended to analyze these issues with relevant citations or examples.
2. The datasets used in the experiments are relatively limited. In comparative methods (e.g., ID-CDM and DCD), experiments were conducted on multiple datasets. It is suggested to include additional datasets for comparative analysis.
3. There is a formatting issue in the paper: the Eq. (15) L_(i,j)^rshould correspond to the definition of L_(i,j)^res in Eq. (9).

**Questions:**

The paper proposes applying probabilistic box embeddings to CD tasks, how the overlapping regions in box embeddings associate with the cognitive states of learners and exercises.

**Reviewer Confidence:**

3: The reviewer is confident but not certain that the evaluation is correct

**Scope:**

3: The work is somewhat relevant to the Web and to the track, and is of narrow interest to a sub-community